# Anti-Inflammatory and Tau Phosphorylation–Inhibitory Effects of Eupatin

**DOI:** 10.3390/molecules25235652

**Published:** 2020-11-30

**Authors:** Ching-Hsuan Chou, Kai-Cheng Hsu, Tony Eight Lin, Chia-Ron Yang

**Affiliations:** 1School of Pharmacy, College of Medicine, National Taiwan University, Taipei 10050, Taiwan; d09423201@ntu.edu.tw; 2Graduate Institute of Cancer Biology and Drug Discovery, College of Medical Science and Technology, Taipei Medical University, Taipei 11031, Taiwan; piki@tmu.edu.tw (K.-C.H.); tonyelin@tmu.edu.tw (T.E.L.); 3Ph.D. Program for Cancer Molecular Biology and Drug Discovery, College of Medical Science and Technology, Taipei Medical University, Taipei 11031, Taiwan; 4Ph.D. Program in Biotechnology Research and Development, College of Pharmacy, Taipei Medical University, Taipei 11031, Taiwan; 5Biomedical Commercialization Center, Taipei Medical University, Taipei 11031, Taiwan; 6Master Program in Graduate Institute of Cancer Biology and Drug Discovery, College of Medical Science and Technology, Taipei Medical University, Taipei 11031, Taiwan

**Keywords:** Alzheimer’s disease, neuroinflammation, eupatin, tau, glycogen synthase kinase 3β

## Abstract

Alzheimer’s disease (AD), which is among the most prevalent neurodegenerative diseases, manifests as increasing memory loss and cognitive decline. Tau phosphorylation and aggregation are strongly linked to neurodegeneration, as well as associated with chronic neuroinflammatory processes. The anti-inflammation effects of natural products have led to wide recognition of their potential for use in treating and preventing AD. This study investigated whether eupatin, a polymethoxyflavonoid found in *Artemisia* species, has inhibitory effects on neuroinflammation and tau phosphorylation. We treated mouse macrophages and microglia cells with lipopolysaccharides (LPSs) to activate inflammatory signals, and we treated neuronal cells with a protein phosphatase 2A inhibitor, okadaic acid (OA), or transfection with pRK5-EGFP-Tau P301L plasmid to induce tau phosphorylation. The results indicated that eupatin significantly reduced the LPS-induced protein expression and phosphorylation of p65 and inducible nitric oxide synthase as well as downstream products interleukin 6 and nitrite, respectively. Furthermore, eupatin markedly inhibited the expression of phospho-tau in response to OA treatment and plasmid transfection. We discovered that this inhibition was achieved through the inhibition of glycogen synthase kinase 3β (GSK3β), and molecular docking results suggested that eupatin can sufficiently bind to the GSK3β active site. Our results demonstrate that eupatin has neuroprotective effects, making it suitable for AD treatment.

## 1. Introduction

Alzheimer’s disease (AD) is a common progressive neurodegenerative disease and constitutes a global health concern. Approximately 50 million people have dementia worldwide, and this number is expected to nearly triple by 2050 [1]. AD causes behavioral changes and continual degradation of mental function, resulting in functional decline in learning capacity [2]. The causes of AD are unknown, but oxidative stress and neuroinflammation have been identified as factors contributing to its pathogenesis [2]. Current drugs approved for AD treatment include acetylcholine esterase inhibitors and the *N*-methyl-d-aspartate receptor antagonist; however, these agents only provide symptomatic relief, and their benefits remain limited [3,4]. Therefore, the development of new drugs for AD is urgently required.

Studies have indicated that extracellular amyloid β (Aβ) plaques and intracellular tau neurofibrillary tangles (NFTs) are two neuropathological hallmarks of AD. Evidence suggests that a crosslink exists between these two markers, although the detailed mechanisms remain unclear [5]. Aβ formation in amyloid precursor protein transgenic mice causes hyperphosphorylation of tau, potentially leading to synaptic dysfunction and ultimately neuronal loss [5]. Tau is a microtubule-associated protein that contributes to microtubule stabilization, neuronal cell growth, axon morphology, and transportation under physiological conditions [5]. However, in pathological diseases, the hyperphosphorylation of tau diminishes its ability to bind and stabilize microtubules, resulting in microtubule sabotage [4,6]. Furthermore, phosphorylated tau not only aggregates to assemble neurotoxic NFTs, thus causing neuronal death (a pathological characteristic of AD), but also creates a vicious cycle with innate immune response [6]. Research has indicated that hyperphosphorylated tau can be secreted extracellularly. Moreover, hyperphosphorylated tau can help in activating microglia and reactivating astrocytes, and release neurotoxic inflammatory molecules or cytokines, including tumor necrosis factor α (TNF-α), interleukin 1β (IL-1β) and IL-6 [7]. A detrimental cycle is perpetuated because the activation of glial cells enhances tau pathology through the process of modulating tau kinases (e.g., glycogen synthase kinase 3β (GSK3β) and cyclin-dependent kinase 5 (cdk5)) [8]. Therefore, the modulation of tau phosphorylation and of neuroinflammation may be promising therapeutic strategies for AD.

For many years, flavonoids have been used in a variety of disease treatments. Their great diversity and distribution class them as a good choice among the available therapeutic agents. As one of the main components used for synthesizing various drugs that also have potential for use with natural products (e.g., modifying functional motifs in natural product skeleton to alter lipophilicity or bioactivity) [9], flavonoids are crucial for discovering new drugs [10]. An increasing body of evidence suggests that cognitive function can be significantly improved through dietary consumption of flavonoids, thus inhibiting or delaying the senescence process related to neurodegenerative disorders such as AD [11]. The mechanisms underlying these changes are unclear, but they may be linked to neuronal apoptosis inhibition resulting from oxidative stress, neuroinflammation, or inhibition of the primary enzymes involved in amyloid plaque and other pathological product formation [12]. However, the detailed mechanism still requires further investigation. *Artemisia annua* L., commonly known as Qinghao, has traditionally been used for treating chills and fever in China. *A. annua* and other *Artemisia* species are traditional components of treatment for febrile conditions, inflammation, and pain [13]. Eupatin, a polymethoxyflavonoid in *Artemisia* species, has been recognized to possess nitric oxide–reducing and acetylcholinesterase-inhibitory effects [14]. However, the inhibition of tau phosphorylation and the underlying mechanism remain unknown. In this study, we evaluated the anti-inflammatory effect of eupatin and further determined whether eupatin can reduce tau phosphorylation to achieve neuroprotective effects against AD.

## 2. Results

### 2.1. Eupatin Exhibited Protent Anti-Inflammatory Activities

The structure of eupatin is displayed in Figure 1A. We first investigated the ability of eupatin to inhibit cell viability in RAW264.7 macrophages and Neuro-2a cells. As displayed in Figure 1B,C, neither 0.5 μM nor 2 μM eupatin treatment altered the cell viability of these two cell types. We also evaluated the cytotoxicity of other flavonoids, namely epigallocatechin-3-gallate (EGCG) and luteolin, in both cell types. Our results showed that neither EGCG (10 and 50 μM) nor luteolin (1 and 10 μM) treatment caused cytotoxicity (Figure 1B,C). Therefore, further studies were conducted at these concentrations unable to induce cell toxicity. NF-κB signals are known regulators of transcription in various inflammatory factors; examples include IL-6 and nitric oxide in macrophages and microglia. We next studied whether eupatin exhibited anti-inflammatory effects to downregulate lipopolysaccharide (LPS)-induced NF-κB–dependent inflammation in macrophages and microglia cells (RAW264.7 and BV-2, respectively).

The results indicated that LPS treatment significantly increased the phosphorylation of p65 (Figure 2A, Appendix A), thereby enhancing the expression of inducible nitric oxide synthase (iNOS; Figure 2A) and IL-6 (Figure 3A) in RAW264.7 macrophages. We also measured the amount of nitrite to determine nitric oxide production. The nitrate level detected after LPS treatment of RAW264.7 cells was significant (Figure 3B). Eupatin significantly attenuated LPS-induced p65 phosphorylation, expression of iNOS (Figure 2A) and IL-6, and nitrate production (Figure 3). EGCG and luteolin were found to have similar inhibitory effects and were used as positive controls. Similar results were obtained for BV2 microglia cells. Eupatin treatment markedly inhibited LPS-triggered p65 phosphorylation, iNOS and COX-2 expression (Figure 2B, Appendix A), and nitrite production in BV2 cells (Figure 3C).

### 2.2. Eupatin Significantly Inhibited Hyperphosphorylation of Tau

Okadaic acid (OA) potently inhibits protein phosphatase 2A (PP2A) and is capable of inducing in vitro and in vivo AD-like tau hyperphosphorylation [15,16]. We observed significant tau phosphorylation in neuronal Neuro-2a cells when using OA to identify the inhibition of tau phosphorylation by eupatin. In this study, 60 nM OA treatment for 6 h significantly increased the expression of tau protein phosphorylated at Ser262, Ser202/Thr205 and Ser396 in Neuro-2a cells but did not increase the total tau level (Figure 4, Appendix A). Eupatin (2 μM) significantly inhibited the phosphorylation of tau at Ser262, Ser202/Thr205 and Ser396 in Neuro-2a cells in response to OA treatment, but the total tau level was unchanged (Figure 4). Moreover, 50 μM EGCG treatment inhibited tau phosphorylation at multiple sites, but the inhibitory effect was not as potent as that of eupatin. Luteolin only mildly inhibited tau phosphorylation at Ser262 (Figure 4).

In addition, Neuro-2a cells transfected with pRK5-EGFP-Tau P301L plasmid, which encodes human mutant P301L-tau, caused tau phosphorylation at multiple sites, including Ser262, Ser202/Thr205 and Ser396 (Figure 5, Appendix A). This model was also used to evaluate the inhibition of tau phosphorylation in eupatin treatment. EGCG and luteolin only achieved mild inhibition at the Ser262 or Ser202/Thr205 site; however, eupatin (2 μM) treatment achieved potent inhibition of tau phosphorylation at all sites (Figure 5). These results demonstrated that eupatin significantly inhibited tau hyperphosphorylation.

### 2.3. Mechanism Studies of Phosphorylated Tau Inhibition in Response to Eupatin Treatment

GSK3β is a major kinase involved in aberrant tau hyperphosphorylation [6]. GSK3β activity is elevated in the brains of patients with AD [17], and overactivation in mice causes tau to be hyperphosphorylated and leads to tau pathology similar to that in patients with AD [18]. The phosphorylation of Tyr216 enhances GSK3β activity [19]. Thus, we identified whether the inhibition of tau phosphorylation by eupatin occurs through the modulation of GSK3β activity. OA treatment significantly increased GSK3β phosphorylation of Tyr216, but no clear changes in GSK3β levels were observed. EGCG and luteolin treatment significantly reduced GSK3β phosphorylation of Tyr216; however, eupatin (2 μM) treatment exhibited the most potent inhibitory effect (Figure 6A, Appendix A). We also performed molecular docking to determine the interactions between eupatin and GSK3β. The docking results revealed that eupatin occupied the GSK3β active site. Eupatin can be separated into two groups, Group 1 and Group 2, which contain a benzopyran ring and a benzene ring, respectively (Figure 6B). The benzopyran and benzene ring systems are typical of flavonoid structures. Group 1 forms two hydrogen bonds with hinge residue V135. This formation process is facilitated by the hydroxyl and carbonyl moieties on the benzopyran ring, which function as a hydrogen donor and acceptor, respectively (Figure 6C). Interactions with hinge residues are a common feature of small molecules that target kinase active sites [20]. Group 2 contains a methoxy moiety on the aromatic ring, which forms an additional hydrogen bond with residue K85 (Figure 6B). A mutation with the lysine residue K85 can produce a “kinase-dead” GSK3β mutant, which suggests that lysine residue K85 is essential for GSK3β kinase activity [21]. Together, these hydrogen bonds anchor eupatin within the GSK3β active site. Additionally, the cyclic rings of eupatin induce several hydrophobic interactions. Group 1 occupies a small hydrophobic pocket formed by residues I62 and V70 (Figure 6C). An additional hydrophobic interaction occurs between the benzopyran ring and the carbon sidechain of residue L188. The aromatic ring of Group 2 produces hydrophobic interactions with a pocket formed by residues A83, K85, and L132. Hydrophobic contact occurs with the side chain of residue L188 (Figure 6C). Together, these interactions suggest that eupatin can sufficiently bind to the GSK3β active site.

## 3. Discussion

Neuroinflammation is known to contribute to the neuronal damage that underpins neurodegenerative disorders such as AD. Several flavonoid compounds have undergone comprehensive investigation and been recognized for their potential for preventing and treating neurodegenerative diseases [11]. The most prevalent and bioactive polyphenol obtained from solid green tea extract is EGCG, which has been reported to possess strong anti-inflammatory properties and protect against brain edema and neuronal injury [22]. EGCG has also been demonstrated to be capable of suppressing TNF-α, IL-1β, IL-6, and iNOS expression, restoring intracellular antioxidant levels for preventing proinflammatory effects in microglia caused by free radicals [22], and suppressing Aβ-induced neurotoxicity by inhibiting GSK3β activation and c-Abl/FE65 nuclear translocation [23]. In vivo results obtained using a transgenic mouse model of AD also indicated that EGCG, when administered orally, inhibits Aβ deposition in the cortex and hippocampus and suppresses tau phosphorylation [24]. Similar results have been reported for luteolin. In the context of AD, one study on mice reported that when used in primary neuronal cells with SweAPP overexpression and in Neuro-2a cells transfected with SweAPP, luteolin significantly reduced Aβ production [25]. A subsequent study reported that luteolin attenuated zinc-induced tau hyperphosphorylation at Ser262/356 in SH-SY5Y cells through its antioxidant effects [26]. Relevant studies have rarely discussed the pharmacological activity of eupatin, and the present study is the first to describe the inhibitory effect of eupatin on tau phosphorylation. Our results indicate that eupatin not only significantly inhibits p-p65 phosphorylation, iNOS and IL-6 expression, and nitrite production in macrophages and microglia but also markedly downregulates the hyperphosphorylation of tau at multiple sites in neuronal cells. The results suggest eupatin possesses the potential to prevent hyperphosphorylated tau-mediated detrimental cycle in AD. The ability of eupatin to cross the blood–brain barrier (BBB) requires further study; however, one study indicated that orally administered eupatilin, a methoxy-derivative of flavonoids, can cross the BBB and significantly reduce brain infraction in mice [27]. Another study indicated the AUC_0–t_ of casticin, a methoxy-derivative in chromone ring of eupatin, the oral administration 400 mg/kg and intravenous injection 50 mg/kg in rat are 18,652.72 ± 4030.88 ng h/mL and 41,225.92 ± 1403.37 ng h/mL, respectively [28]. The mean plasma concentration of casticin can be calculated as 14.24 ± 3.08 μM and 110.13 ± 3.75 μM, respectively. Comparing these concentration, 2 μM eupatin would be rationally reached in vivo. These results suggest that eupatin has potential for use in AD treatment.

Tau is known to play a key role in microtubule assembly and stabilization. One study indicated that phosphorylation is central to the regulation of tau’s physiological functions, such as microtubule binding, and is thus responsible for regulating microtubule stabilization and assembly [6]. Research has suggested that the phosphorylation sites Ser202, Thr205, and Ser262 are involved in tau conformational change, that Ser262 and Ser356 are involved in tau dissociation from microtubules, and that Thr231, Ser396, and Ser404 are involved in tau aggregation [6,29]. Therefore, tau phosphorylation inhibitors, protein phosphatases activators, tau aggregation inhibitors, and phosphorylated tau eliminators are considered to have therapeutic potential [30]. GSK3β is a major kinase involved in aberrant tau phosphorylation. Because GSK3β has inherently high activity, inhibiting and activating phosphorylation on Ser9 and Tyr216, respectively, are its primary means of regulation [19]. Furthermore, when overexpressed in mice, GSK3β causes tau hyperphosphorylation and leads to tau pathology similar to that in patients with AD [16,31]. Phosphorylation in regions bordering the microtubule-binding domain (the primary sites of GSK3β-mediated tau phosphorylation) has been noted to lead to tau detaching from microtubules, resulting in self-aggregation [32]. Furthermore, as well as kinase, phosphatase contributes to the phosphorylation of tau. Patients with AD have reduced tau phosphatase activity, most of which is caused by PP2A [6]. OA has been identified as a potent PP2A activity inhibitor that can induce AD-like tau hyperphosphorylation in vitro and in vivo [16,33]. Our prior research revealed the occurrence of significant tau phosphorylation in Neuro-2a cells that were given OA or transfected with the pRK5-EGFP-Tau P301L plasmid, which encodes human mutant P301L-tau [32]. The results of the present study revealed that eupatin significantly inhibited OA- or plasmid-induced tau phosphorylation at multiple sites. This inhibition occurred primarily as a result of binding to GSK3β and the downregulation of enzyme activity.

## 4. Materials and Methods

### 4.1. Materials

Eupatin was obtained from the National Cancer Institute (NCI No. 122412). We purchased primary antibodies against p-GSK3β (Tye216) and p-p65 (Ser536) from Cell Signaling Technology (Danvers, MA, USA). We obtained antibodies against p-tau (Ser396) from Abcam (Cambridge, MA, USA) and those against p-tau (Ser202/Thr205), p-tau (Ser262), and tau w from Thermo Fisher Scientific (Waltham, MA, USA). Antibodies against GSK3β and α-tubulin were obtained from GeneTex Inc. (Hsinchu, Taiwan), and antibody against iNOS was purchased from Santa Cruz Biotechnology (Santa Cruz, CA, USA). Jackson ImmunoResearch Inc. (West Grove, PA, USA) was our source for horseradish peroxidase (HRP)-conjugated antimouse and antirabbit immunoglobulin G secondary antibodies, and OA was purchased from Cayman Chemical Company (Ann Arbor, MI, USA). Unless otherwise stated, we obtained all chemicals not mentioned here from Sigma-Aldrich (St. Louis, MO, USA).

### 4.2. Cell Culture

Murine RAW264.7 cells, a mouse macrophage cell line, were obtained from the Bioresource Collection and Research Center (Hsinchu city, Taiwan). Mouse BV-2 microglia were kindly provided by Prof. Shiow-Lin Pan (Graduate Institute of Cancer Biology and Drug Discovery, Taipei Medical University, Taipei, Taiwan). Dulbecco’s modified Eagle’s medium (DMEM) (Invitrogen Life Technologies, Carlsbad, CA, USA) with supplementation of 100 μg/mL streptomycin (Biological Industries, Kibbutz Beit Haemek, Israel), 10% (v/v) fetal bovine serum (Invitrogen Life Technologies, Carlsbad, CA, USA), and 100 U/mL penicillin was used for cell culturing. We obtained the murine neuroblastoma cell line (Neuro-2a) from the Bioresource Collection and Research Center. After culturing the cells in minimum essential media consisting of streptomycin (100 μg/mL), penicillin (100 units/mL), and 10% fetal bovine serum, we incubated them at a temperature of 37 °C in a humidified 5% CO_2_ air atmosphere.

### 4.3. Cell Cytotoxicity Assay

Colorimetric MTT assay was employed for measuring cell cytotoxicity. For 48 h, 1 × 10^4^ cells were incubated in 96-well plates in 1 mL of medium with the control vehicle or vehicle containing the experimental compound. The next step after the application of different treatments was the addition of MTT (1 mg/mL) and incubation of the plates at 37 °C for another 2 h. We then pelleted and lysed the cells in 10% sodium dodecyl sulfate (SDS) with 0.01 M HCl and used a microplate reader to measure the absorbance at 570 nm.

### 4.4. Nitrate Assay

Nitrite production was measured in RAW264.7 macrophage and BV-2 microglia cell supernatants. In brief, after culturing 1 × 10^6^ cells in 6-well plates and stimulating them using 100 ng/mL LPSs for 24 h, we mixed a Griess reagent with the cell supernatant (100 μL each) and measured the optical density at 550 nm. A standard curve based on obtained concentrations of sodium nitrite dissolved in DMEM was generated for calculating the nitrite concentration.

### 4.5. IL-6 Enzyme-linked Immunosorbent Assay

Thirty-minute incubation of the cells with the experimental compounds was applied prior to and during 24-h incubation with LPSs (100 ng/mL). After collection of the medium, an enzyme-linked immunosorbent assay kit (PeproTech Inc., Cranbury, NJ, USA) was used to perform an assay for IL-6.

### 4.6. Transfection Assay

Cells were seeded 1 day before transfection. After 20 min of mixing 1 μL of TurboFect transfection reagent and pRK5-EGFP-Tau P301L plasmids (1 μg) at room temperature and addition of the mixture to the cells, the suspensions underwent 24-h incubation at 37 °C in a humidified 5% CO_2_ atmosphere.

### 4.7. Immunoblot Analyses

For 10 min at 4 °C, 1 × 10^6^ cells were incubated in lysis buffer (20 mM HEPES, pH 7.4; 2 mM EGTA, 0.1% Triton X-100; 50 mM β-glycerophosphate; 1 mM DTT; 10% glycerol; 1 μg/mL leupeptin; 1 mM sodium orthovanadate; 1 mM phenylmethylsulfonyl fluoride, and 5 μg/mL aprotinin). Next, the cells were removed, placed on ice for 10 min, and subjected to 30-min centrifugation (17,000× *g*) at 4 °C. Next, we electrophoresed 20-μg protein samples on SDS polyacrylamide gels before transferring them onto a nitrocellulose membrane. The nitrocellulose membrane was subsequently blocked through 30-min incubation with 5% bovine serum albumin in Tris-buffered saline containing 0.1% Tween 20 (TBST) at room temperature. Immunoblots were obtained through incubation overnight at 4 °C with primary antibodies in TBST and subsequent 1-h incubation at room temperature with secondary antibodies conjugated with HRP. Measurement of antibody binding was performed through photographic film exposure and application of an enhanced chemiluminescence reagent (GE Healthcare Corp., Buckinghamshire, UK).

### 4.8. Bioinformatics and Protein Modeling

Molecular docking analysis was performed using the Maestro docking module (Schrödinger). The GSK3β crystal structure was obtained from the Protein Data Bank (PDB ID: 1UV5) [34]. The cocrystal ligand (HET ID: BRW) was used to identify the active site and define the center of the receptor grid. The compound (eupatin) was prepared using LigPrep. The ionization state was generated at a pH of 7.0 ± 2.0 by using the Epik module in LigPrep. The compound was docked to the active site using the Glide module of Schrödinger Maestro. A three-dimensional image of the docked compound and the GSK3β active site was generated using PyMOL (PyMOL Molecular Graphics System, Schrödinger LLC, New York, NY, USA), and a two-dimensional interaction diagram was generated using Maestro.

### 4.9. Data Analysis and Statistics

One-way analysis of variance was used for analyses, and data are presented as mean ± standard error of the mean. When significant differences were detected between groups, Tukey’s post hoc test was applied for identifying the significantly different group pairs. The significance level for parameters was set at *p* < 0.05.

## 5. Conclusions

Eupatin treatment significantly inhibited LPS-triggered p65 activation and inflammatory factors production, and markedly reduced tau phosphorylation on Ser262, Ser202/Thr205, and Ser396. Our novel findings indicate that eupatin has neuroprotective potential against AD.

## Figures and Tables

**Figure 1 molecules-25-05652-f001:**
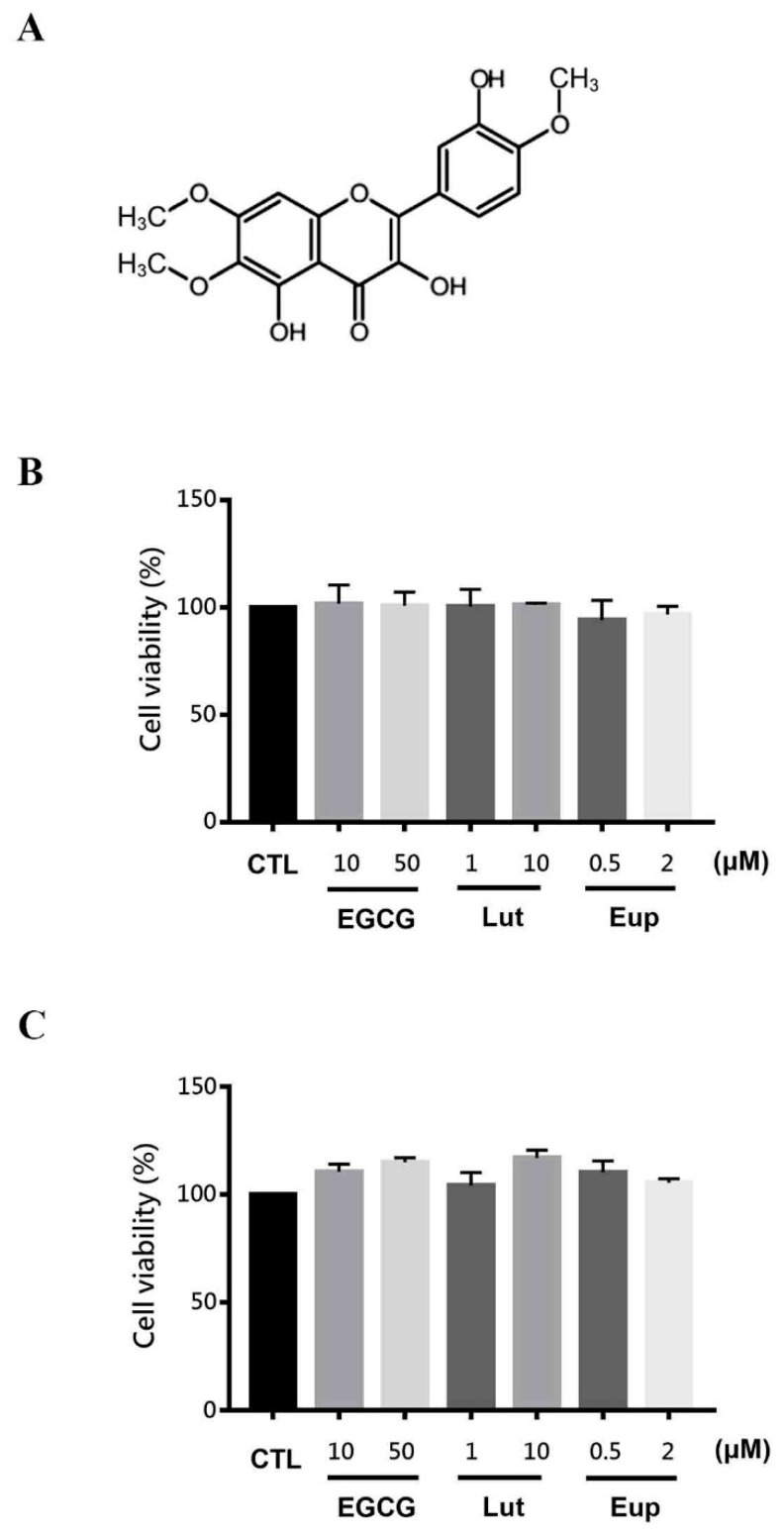
Eupatin cell viability. (**A**) Eupatin chemical structure. (**B**,**C**) 24-h incubation of (**B**) RAW264.7 and (**C**) Neuro-2a cells with or without the specified eupatin, luteolin, and EGCG concentrations. MTT assay was employed for determining cell viability. The results are presented as the mean ± standard error of the mean of three independent experiments.

**Figure 2 molecules-25-05652-f002:**
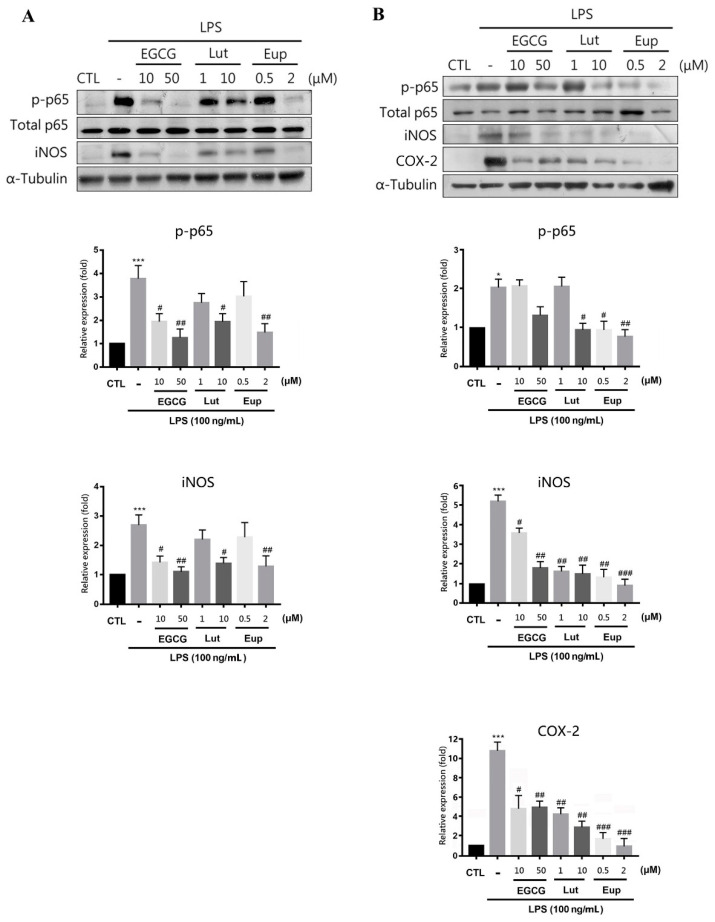
Eupatin significantly inhibited the phosphorylation of NF-κB, iNOS, and COX-2 expression in lipopolysaccharide (LPS)-activated macrophages and microglia. (**A**) RAW264.7 and (**B**) BV-2 cells were treated with indicated concentrations of epigallocatechin-3-gallate (EGCG), luteolin, or eupatin for 30 min and then stimulated with LPSs (100 ng/mL) for 24 h. Western blotting of the cell lysates was performed with the indicated antibodies. The results are presented as the mean ± standard error of the mean of three independent experiments. * *p* < 0.05 and *** *p* < 0.001 for comparisons with the control group; # *p* < 0.05, ## *p* < 0.01, and ### *p* < 0.001 for comparisons with the group treated with LPSs.

**Figure 3 molecules-25-05652-f003:**
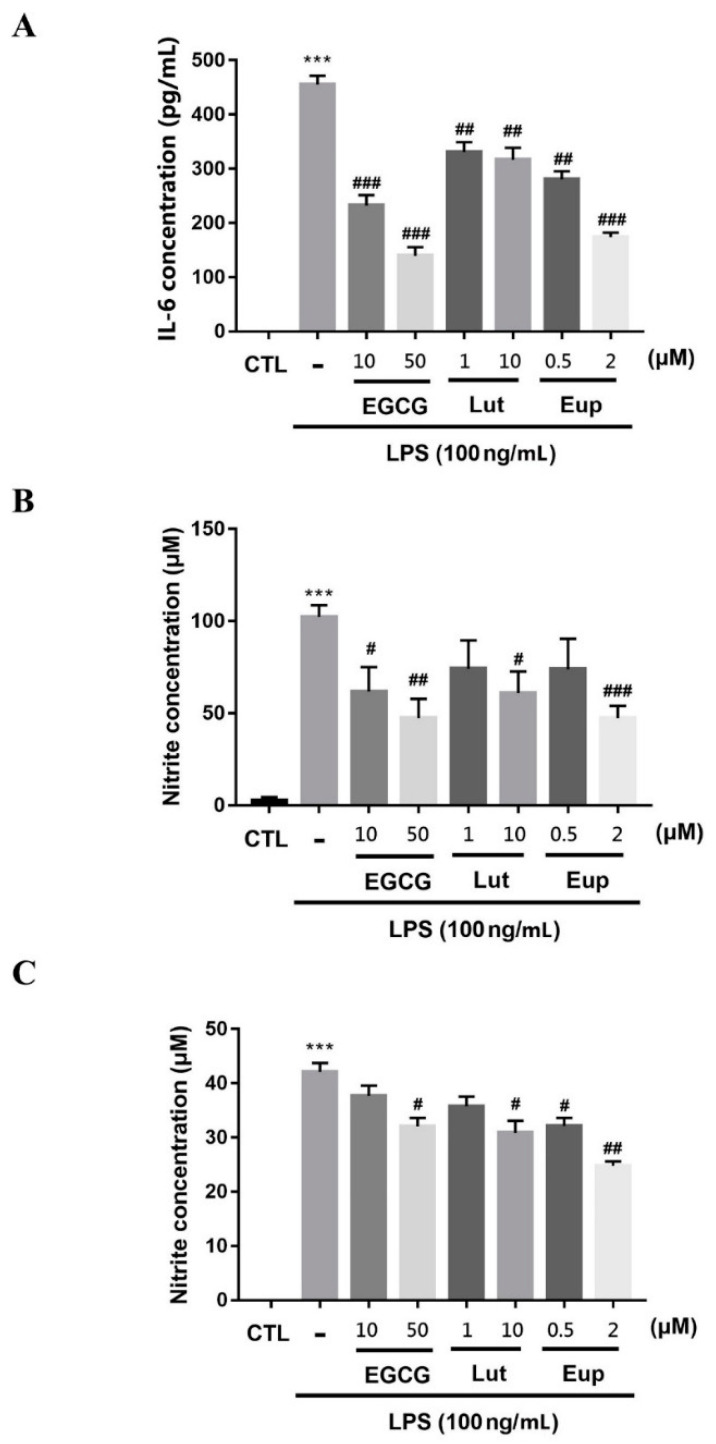
Eupatin’s ability to inhibit LPS-induced production of IL-6 and nitric oxide. After 30-min incubation of (**A**,**B**) RAW264.7 and (**C**) BV-2 cells with the indicated concentrations of EGCG, luteolin, or eupatin, the cells were stimulated for 24 h using 100 ng/mL LPSs. The supernatants were subsequently collected and assayed for IL-6 and nitrite. The results are presented as the mean ± standard error of the mean of three independent experiments. *** *p* < 0.001 for comparisons with the control group; # *p* < 0.05, ## *p* < 0.01, and ### *p* < 0.001 for comparisons with the LPS-treated group.

**Figure 4 molecules-25-05652-f004:**
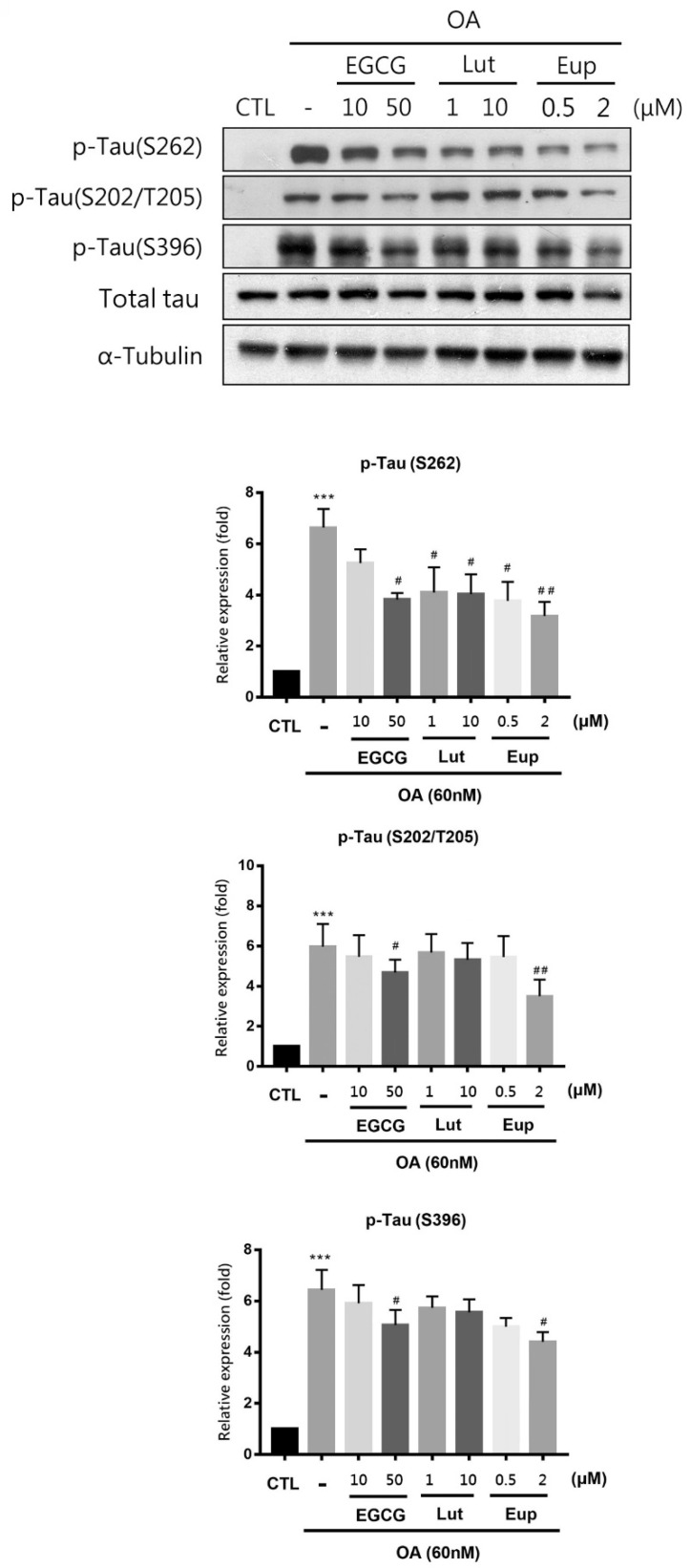
Eupatin significantly inhibited okadaic acid (OA)-induced tau phosphorylation. Neuro-2a cells were treated with indicated concentrations of EGCG, luteolin, or eupatin for 24 h before 6-h incubation with 60 nM OA. Western blotting of cell lysates was performed using the specified antibodies. The results are presented as the mean ± standard error of the mean of three independent experiments. *** *p* < 0.001 for comparisons with the control group; # *p*  <  0.05 and ## *p*  <  0.01 compared with the OA-treated group.

**Figure 5 molecules-25-05652-f005:**
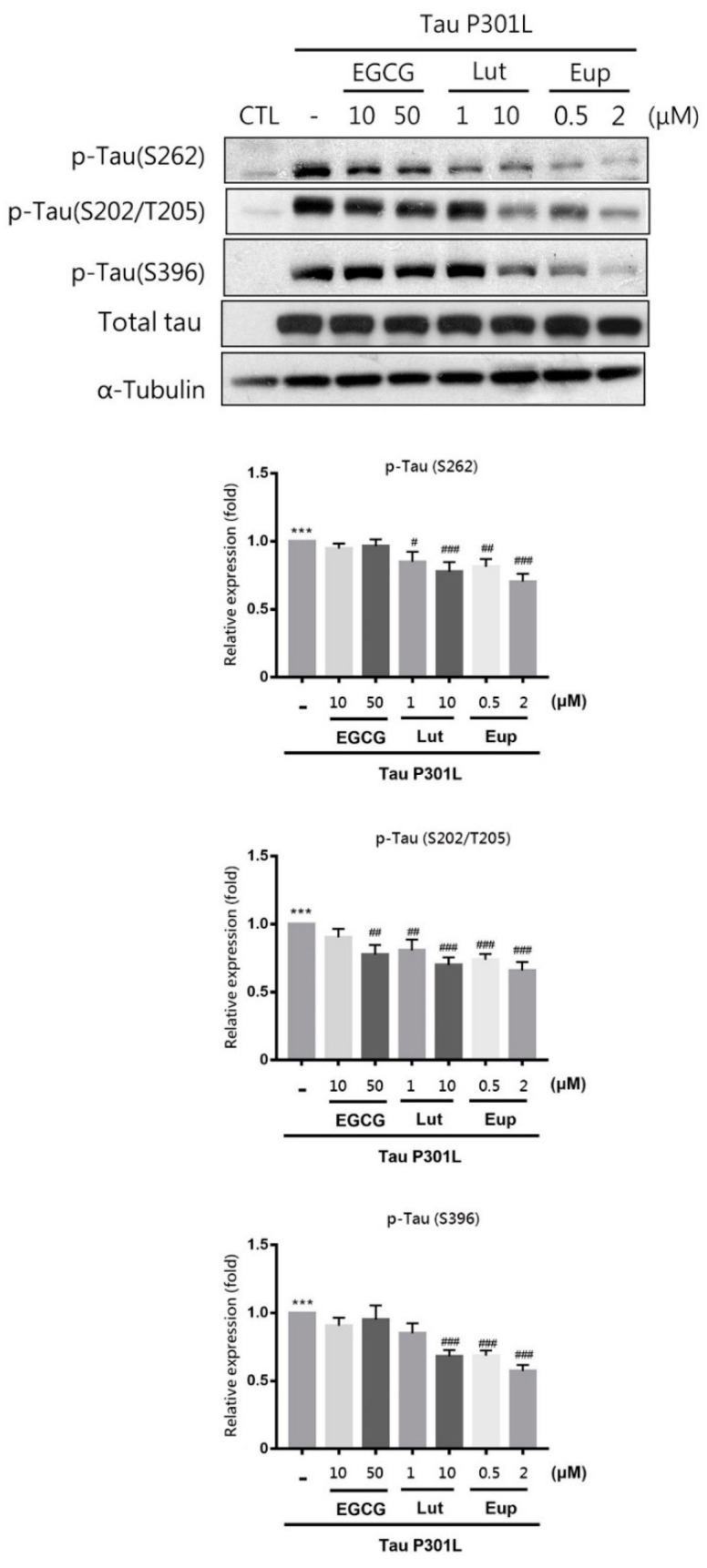
Eupatin achieved significant inhibition of tau phosphorylation. Neuro-2a cells underwent 24-h transfection using pRK5-EGFP-Tau P301L plasmid and were subsequently incubated with or without the indicated concentrations of EGCG, luteolin, or eupatin for 24 h. Western blotting of cell lysates was performed using the indicated antibodies. The results are presented as the mean ± standard error of the mean of three independent experiments. *** *p* < 0.001 for comparisons with the control group; # *p*  <  0.05, ## *p*  <  0.01, and ### *p*  <  0.001 for comparisons with the plasmid transfection group.

**Figure 6 molecules-25-05652-f006:**
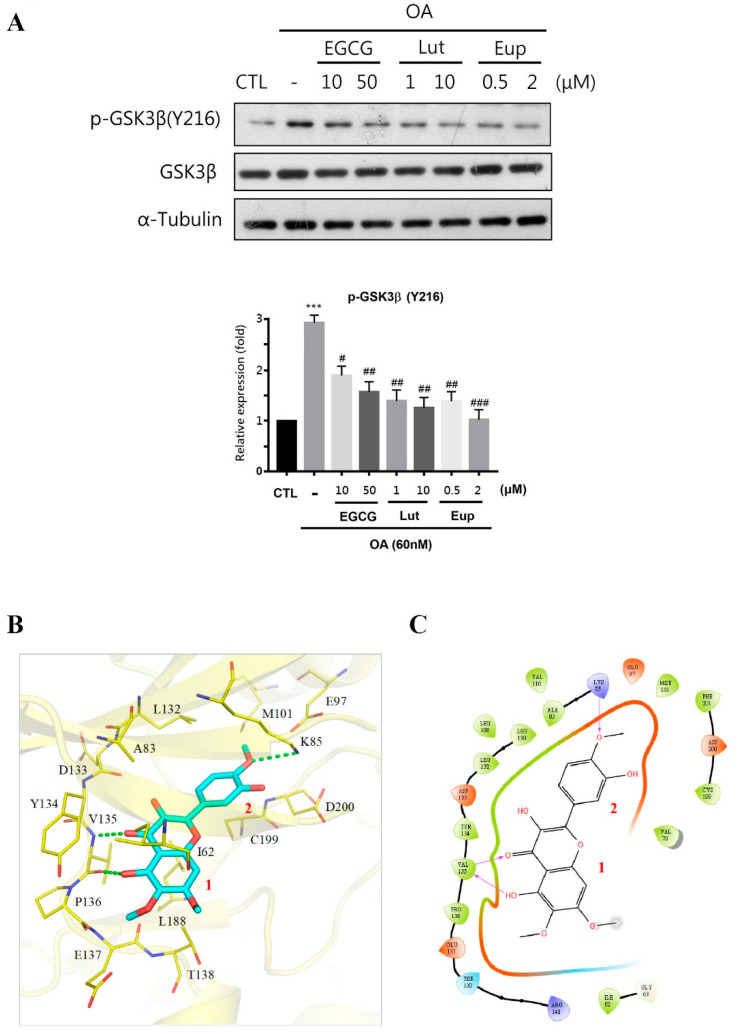
Eupatin significantly inhibited GSK3β activation. (**A**) Neuro-2a cells were incubated with indicated concentrations of EGCG, luteolin, or eupatin for 24 h and subsequently underwent 6-h incubation with 60 nM OA. Western blotting of cell lysates was performed using the indicated proteins. The results are presented as the mean ± standard error of the mean of three independent experiments. *** *p* < 0.001 for comparisons with the control group; # *p*  <  0.05, ## *p*  <  0.01, and ### *p*  <  0.001 for comparisons with the OA-treated group. (**B**) The docking pose of eupatin (blue) in the GSK3β active site (yellow). Hydrogen bonds are indicated by dashed green lines, and active site residues are labeled. The benzopyran and benzene rings are labeled as Groups 1 and 2, respectively. (**C**) Two-dimensional interaction diagram of eupatin and GSK3β. Arrows denote hydrogen bonds, and residues in green indicate hydrophobic interactions.

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
