# Peer review of "Anti-Inflammatory and Tau Phosphorylation–Inhibitory Effects of Eupatin"

_molecules, 2020, doi:10.3390/molecules25235652_

Round 1

Reviewer 1 Report

Anti-inflammatory and Tau Phosphorylation– 3 inhibitory Effects of Eupatin

This manuscript investigated whether  eupatin, a polymethoxy-flavonoid found in Artemisia species, has inhibitory effects on neuroinflammation and tau phosphorylation.

The results indicated that eupatin significantly reduced the LPS-induced protein expression and phosphorylation of p65 and inducible nitric oxide  synthase as well as downstream products interleukin and nitrite, respectively.

The autors discovered that this inhibition was achieved through the inhibition of glycogen  synthase kinase 3β (GSK3b), and molecular docking results suggested that eupatin can sufficiently  bind to the GSK3b active site. Our results demonstrate that eupatin has neuroprotective effects, making it suitable for AD treatment.

Furthermore, phosphorylated tau not only aggregates to assemble neurotoxic NFTs, thus causing neuronal death (a pathological characteristic of AD), but also creates a vicious cycle with innate immune response.

Eupatin treatment significantly inhibited LPS-triggered p65 activation and inflammatory factors  production, and markedly reduced tau phosphorylation on Ser262, Ser202/Thr205, and Ser396. Our novel findings indicate that eupatin has neuroprotective potential against AD.

The Chapter Results and Discussion is of good scientific quality and the rich and instructive graphic realizes the understanding of the obtained results and of their significance. The experimental data is described appropriately and the manuscript needs no language and grammar corrections. The manuscript is written straight forward.

The study is a meaningful suppliment  to the series of publications regarding the heterocyclic and macrocyclic compounds (with P, S, N atoms) related to natural products, that  have been extensively studied because their important properties and applications, especially in biological activities, such as, anti-microbial, anti-proliferative (prostate cancer cells),  anti-cancer , anti-influenza A, neuraminidase (NA) , and with antioxidant activity.

In Introduction the autors have been described that  hyper-phosphorylated tau can help in activating microglia and reactivating astrocytes, and release neurotoxic inflammatory molecules or cytokines, including tumor necrosis factor a (TNF-α), interleukin 1b (IL-1β) and IL-6.

Because the authors have been presented in the References part of the manuscript a series of scientific papers,   that  hyper-phosphorylated tau can help in  release neurotoxic inflammatory molecules or cytokines, including tumor necrosis factor a (TNF-α), interleukin 1b (IL-1β) and IL-6, the authors have also to present in the introduction part the data about the other    phosphorylated derivatives,  heterocyclic and macrocyclic compounds (with P, S, N atoms) related to natural pro-ducts,  with the important applications as  receptors especially in biological activities.

Examples of relevant publications are given below. It is recommended to the authors to cite these papers to give their introduction a wider base.

  • Trifluoromethylpyridine-Substituted N-Heterocyclic Carbenes (NHC) related to Natural Products: Synthesis, Structure and Potential Antitumor Activity of some Corresponding Gold(I), Rhodium(I) and Iridium(I) Complexes.
  • Elena Maftei, Catalin V. Maftei, Peter G. Jones, Matthias Freytag1, Heiko Franz, Gerhard Kelter, Heinz-Herbert Fiebig, Matthias Tamm* and Ion Neda* , Helv. Chim. Acta, 2016, 99, 469-481     
  • N-Heterocyclic Carbenes (NHC) Derived from Imidazo[1,5-a]pyridines Related to Natural Products: Synthesis, Structure and Potential Biological Activity of Some Corresponding Gold(I) and Silver(I) Complexes.
  •  
  • Monica Mihorianu, M. Heiko Franz2,4, Peter G. Jones, Matthias Freytag1 Gerhard Kelter, Heinz-Herbert Fiebig, Matthias Tamm*and Ion Neda *
  • Organometal. Chem. 2016, 30, 581-589
  •  
  • Novel 1,2,4-oxadiazoles and trifluoromethylpyridines related to natural products: Synthesis, structural analysis and investigation of their antitumor activity(Article)
  • Maftei, C.V., Fodor, E., Jones, P.G., Daniliuc, C.G., Franz, M.H., Kelter, G., Fiebig, H.-H., Tamm, M. , Neda, I., Tetrahedron, 2016,72,1185-1199

Some references should be inserted.

In conclusion of my review,

I recommend this manuscript for publication with minor revisions!

Author Response

We provide a point-by-point response to reviewer's comments as PDF file.

Reviewer 2 Report

In the paper "Anti-inflammatory and inhibitory effects of eupatin Tau phosphorylation" by Ching-Hsuan et al, the authors tested the effect of eupatin, a polymethoxyflavonoid, on different cell lines and charcaterized its molecular mechanisms of action. They found that eupatin exherts antiinflammatory properties in in microglia and macrophages (reduction of LPS-induced p65 protein expression and phosphorylation, decrease iNOS). Furthermore, the molecule is able to prevent tau phosphorylation in response to OA treatment, in Neuro-2a cells. In neuronal cells, mechanicistic studies suggest that eupatin binds to the GSK3β active site and modulates its kinase activity. Findings may have implications in AD prevention and treatment.

Eupatin is a new molecule and its properties are quite interesting and very impressive. However, I have some concerns about data presentation. The authors test the molecules in different cellular models involved in AD pathogenesis (important), but the characterization of the Eupatin mechanism of action is only partial.

Main points.

  • Three completely different cell lines were employed in the paper. The authors should better justify why they have chosen these model
  • The idea of the vicious cycle between tau production and inflammation is interesting and should be better developed in the paper. For example, can eupatin prevent the P-tau-mediated activation of microglia and macrophages?
  • Eupatin concentration. The authors propose eupatin for AD treatment and prevention. Very little is known about eupatin in the literature. Can Eupatin exerts toxic effects at higher concentrations? Could, 2 microM concentration, be rationally reached in blood? and in the CNS?
  • Authors should better discuss the effective use of eupatin in vivo (absorption, degradation, bioavailability, etc.)
  • Western blotting images only report the section corresponding to the proteins of interest. The whole blot should be shown in the supplementary data.
  • The text would benefit from English revision.

Author Response

We provide a point-by-point response to the reviewer's comments as PDF file.
